# Intestinal IgA-Coated Bacteria in Healthy- and Altered-Microbiomes (Dysbiosis) and Predictive Value in Successful Fecal Microbiota Transplantation

**DOI:** 10.3390/microorganisms11010093

**Published:** 2022-12-29

**Authors:** Herbert L. DuPont, Zhi-Dong Jiang, Ashley S. Alexander, Andrew W. DuPont, Eric L. Brown

**Affiliations:** 1Center for Infectious Diseases, Division of Epidemiology, Human Genetics and Environmental Sciences, University of Texas School of Public Health, Houston, TX 77030, USA; 2Department of Internal Medicine, University of Texas McGovern Medical School, Houston, TX 77030, USA; 3Kelsey Research Foundation, Houston, TX 77005, USA

**Keywords:** secretory IgA, IgA-coated bacteria, IgA-biome, IgA-Seq dysbiosis, fecal microbiota transplantation

## Abstract

IgA-coated bacteria in the gut (IgA-biome) provide a homeostatic function in healthy people through inhibition of microbial invaders and by protecting the epithelial monolayer of the gut. The laboratory methods used to detect this group of bacteria require flow cytometry and DNA sequencing (IgA-Seq). With dysbiosis (reduced diversity of the microbiome), the IgA-biome also is impaired. In the presence of enteric infection, oral vaccines, or an intestinal inflammatory disorder, the IgA-biome focuses on the pathogenic bacteria or foreign antigens, while in other chronic diseases associated with dysbiosis, the IgA-biome is reduced in capacity. Fecal microbiota transplantation (FMT), the use of fecal product from well-screened, healthy donors administered to patients with dysbiosis, has been successful in engrafting the intestine with healthy microbiota and metabolites leading to improve health. Through FMT, IgA-coated bacteria have been transferred to recipients retaining their immune coating. The IgA-biome should be evaluated in FMT studies as these mucosal-associated bacteria are more likely to be associated with successful transplantation than free luminal organisms. Studies of the microbiome pre- and post-FMT should employ metagenomic methods that identify bacteria at least at the species level to better identify organisms of interest while allowing comparisons of microbiota data between studies.

## 1. Introduction and Methods

The knowledge that up to one-third of bacteria in the gut are coated with secretory IgA (IgA) is not a new concept. The biologic significance of IgA-coated bacteria, their role in a healthy and a diseased microbiome, and their value in predicting successful fecal microbiota transplantation (FMT) are topics of current investigation.

In this review, we will briefly consider the dynamics involved with IgA-coating of intestinal bacteria and the functions of this component of the microbiome. Then, we will examine what is known about IgA-coated bacteria in disease states and dysbiosis (reduction in diversity of the colonic microbiome) and consider what is known about these strains in reestablishing microbiome diversity through FMT. We provide the available evidence that in people with a diverse microbiome, the subset of fecal IgA-coated bacteria (IgA-biome) represent core microbiota associated with gut health and should be studied in patients before and after FMT for their predictive value in successful transplantation. 

In performing this review, the authors examined the literature by searching in PubMed for the following: immune microbiome, IgA AND microbiome, IgA coated intestinal microbiota, IgA AND inflammatory bowel disease, IgA AND dysbiosis, IgA AND inflammation. Importantly publications reviewed identified other articles for review. It is the authors intent to provide researchers interested in the microbiome with a current perspective on the IgA-biome and a bibliography on the topic.

## 2. Origin and Production of Intestinal IgA

The origin of intestinal IgA is in Peyer’s patches lamina propria, and other gut-associated mucosal lymphoid tissue (GALT) where B cells undergo class-switch and affinity maturation with production of IgA-producing plasma cells (PCs) that generate one of two secretory IgA subclasses, IgA1 and IgA2 [1]. Production of IgA is influenced by both dendritic cells and epithelial cells [2], as part of a non-inflammatory process [3]. Dendritic cells widely distributed beneath the epithelial lining are specialized antigen-presenting cells that orchestrate immune response [4] or mediate tolerance [5] to bacterial antigens draining to lymphoid tissues. The IgA-coating of bacteria involves both T cell dependent and T cell independent pathways [6].

## 3. The Immune Microbiome

The surface of the gastrointestinal tract is large, with estimates ranging from 32 m^2^ to 300 m^2^, supporting the colonization of up to 10^14^ total fungi, viruses and bacteria including commensal organisms of low virulence as well as potentially virulent strains with disease-producing capabilities. Barriers preventing the translocation of gut organisms through the mucosa with the potential of causing extraintestinal infections, include a monolayer of epithelial cells, locally produced antimicrobial peptides, an intestinal mucus layer of variable thickness and the intestinal immune system that includes IgA, the only acquired immune component acting directly on the gut microbiota. 

IgM and IgG coating of microbiota can occur in the intestine but at levels significantly lower than that of IgA, with IgM being translocated across the gut epithelium in the same mechanism used to translocate IgA [7,8]. In IgA deficiency, IgM binds to intestinal bacteria and serves a similar function as IgA [9], reacting to gut bacteria more broadly and with only a fraction of the overall homeostatic value as seen for IgA [10]. The source of the IgG that coats bacteria in the gut is likely from extraintestinal regions of the body, gaining access to the intestinal lumen because of a leaky gut [7]. IgG responses to coliform bacteria are most common in inflammatory conditions including, Crohn’s disease and diabetes [11,12,13]. 

Eighty percent of the body’s IgA-producing plasma cells (IgA-PCs) are located in the intestinal mucosa [1,14], with more IgA being produced in the body than all other antibody isotypes combined [15]. In healthy people approximately one-third of gut microbiota are coated with IgA, while with an intestinal inflammatory condition, the proportion can increase up to 70% [16]. IgA-coating potentially occurs for all microbial forms in the intestine, bacteria, viruses and fungi. Most studies have examined IgA-coated bacteria and because of data availability, these studies will be reviewed. 

Nearly all the bacterial constituents of the colon are obligate anaerobes. The concentration of bacteria in the small bowel range from approximately 10^4–5^ per g of content in the duodenum, to 10^7–8^ in the distal ileum where transit is slower [17], compared with 10^11^–10^12^ per g in the colon with even slower propulsive motility [18]. The growth of colonic mucosal biopsy tissue is in the range of 10^7^–10^8^ bacteria per g [19].

Bacteria in the small bowel are less diverse than in the large bowel. The ratio of facultative to strictly anaerobic bacteria decreases as we move from proximal to distal regions of the intestine [17]. Composition of bacteria vary as well with strains of Streptococcaceae, Actinomycetaceae, Actinomycinaceae and Corynebacteriaceae commonly found in the normal healthy small intestine, and strains belonging to Firmicutes, Bacteroidota, and Actinomycetaceae more commonly found in the colon [20]. With dysbiosis in the presence of inflammation of the gut (e.g., *Clostridioides difficile* infection and inflammatory bowel disease), strains of proinflammatory Enterobacteriaceae become prevalent in the small and large intestines [20,21]. 

The core intestinal bacteria strains bound to IgA found in two studies of healthy people belong to the following bacterial families: Steptococcaceae, Nocardiaceae, Bacteroidaceae, Sphingomonadaceae, ud-Sphingomonadales, Comamonadaceae, Pseudomonadaceae, Enterococcaceae, Ruminococcaceae, and Lachnospiraceae [16] and Bacteroideaceae (*Bacteroides*), Bifidobacteriaceae (*Bifidobacterium*), Clostridaceae (*Clostridium*), Enterobacteriaceae, Lachnospiraceae (*Dorea, Blautia* and *Roseburia*), Ruminococcaceae (*Ruminococcus*), and Verrucomicrobiaceae (*Akkermansia*) [22].

The laboratory method of detecting IgA-coated bacteria involves sorting bound bacteria from fecal samples by flow cytometry and employing amplicon sequencing methods to determine taxonomy with the method referred to as IgA-Seq methods [23].

## 4. Factors Important in IgA Coating of Intestinal Bacteria

Known factors involved in immune coating of bacteria in the gut include the location of the organisms in the small versus large bowel and the presence of bacterial strains adjacent to the mucosa rather than in the free lumen. Bacterial coating preferentially occurs in the small bowel because this is where most IgA-PCs are located, and the mucus layer protecting the gut epithelial cell monolayer is thin. There are fewer IgA-PCs in the colon, and the mucus layer is thicker. While mucosal-associated bacteria have not been formally studied for IgA-coating, it is likely the frequency of coating would be higher than for organisms in the gut lumen. The mucosal-associated bacteria are more likely to play a role in modulating the immune system and contributing to microbial signals connecting with the enteric nervous system. 

## 5. Targets of IgA Coating on Bacteria

The IgA released by plasma cells target multiple microbial antigens, including lipopolysaccharide [24], proteins [25] and capsular polysaccharides [26], flagellin and virulence properties of pathogenic microbes such as exotoxins, in a polyreactive response cutting across species [27], with the capacity of reducing organism virulence [27,28,29,30].

## 6. Function of IgA Coated Bacteria

In otherwise healthy people, IgA-bound bacteria are associated with overall gut microbiome health [31]. While the functions of IgA-coated bacteria or viruses are not completely understood, much of the data available comes from studies in mouse models using monoclonal antibodies. The potential ways the IgA-biome benefits its host, identified in either human or animal studies, are included in Figure 1. In certain disease settings, as discussed later, IgA-coated bacteria are associated with gut inflammation.

IgA may facilitate immune inclusion where binding to bacterial polysaccharide can lead to clumping of groups of bacteria with subsequent confinement to the mucus layer of the intestine for biologic value to the host or to sequester the strains away from microbial competitors. IgA binds to virulent enteric bacteria or viruses to protect the gut lining from microbiota by blocking attachment to the epithelium and subsequent translocation across the mucosa in a process referred to as immune exclusion. Many organisms with potential virulence properties, adhesiveness, exotoxin production or potential for causing inflammation are eliminated through this process. IgA is able to enhance fitness and optimize growth of desirable strains. Bound IgA can affect microbial metabolism and gene expression. Motile organisms may be rendered non-motile by IgA binding to bacterial flagellin.

## 7. Immunoglobulin (Ig)-Coating of Bacteria in the Gut in Conditions with Reduced Microbiota Diversity (Dysbiosis), Conditions Potentially Modified by Microbial Restoration with Fecal Microbiota Transplantation

There are limited studies of the IgA-biome in health and in disordered microbiomes or dysbiosis. This section summarizes the conditions associated with dysbiosis where: alteration of the microbiome occurs and IgA-coating of intestinal bacteria has been studied to see how it influences disease progression and response to FMT.

### 7.1. Inflammatory Bowel Disease (IBD)

IBD is listed first because there are more papers dedicated to the study of the IgA-biome for these conditions than for others. 

In a study of 175 patients with Crohn’s disease (CD), 75 with ulcerative colitis (UC) and 41 healthy controls, IgA- and IgG-coated bacteria were measured along with other clinical aspects of disease [32]. In the group with CD, the percent of IgA- and IgG-coated bacteria were positively, significantly, and directly correlated with markers of inflammation, C-reactive protein (CRP) and erythrocyte sedimentation rate (ESR) (*p* < 0.0001 for both), and severity of disease as measured by Crohn’s disease activity index (CDAI) (*p* < 0.0001 for both immunoglobulins), and Simple Endoscopic Score in Crohn’s disease (SES-CD) (*p* < 0.0001 for both immunoglobulins). For UC, the percentage of IgA-coated bacteria increased in left-sided colitis, while in extensive colitis, a higher percentage of IgG-coated bacteria were found compared with healthy controls. Other correlations in UC with percentage of IgA-or IgG were with CRP and ESR (*p* < 0.0001 for IgA-coating and *p* < 0.0002 for IgG-coating), Mayo Score (*p* < 0.0001 for IgA-coated and 0.0006 for IgG-coated bacteria) and Ulcerative Colitis Endoscopic Index of Severity (UCEIS) (*p* < 0.0001 for IgA-coated bacteria and *p* = 0.0002 for IgG-coated bacteria). Not only was the percent of IgA-coated bacteria increased in the two forms of inflammatory bowel disease, but the concentrations of soluble IgA and IgG in feces were shown to be significantly higher in patients with IBD compared with healthy controls. These levels were directly related to severity of disease.

In a second study, 42 patients with IBD (22 with CD and 20 with UC) and 12 healthy controls were studied for IgA-, IgG- and IgM-coated bacteria [33]. Both forms of IBD showed higher proportions of IgA-coated bacteria. The proportion of all Ig subtype-coated bacteria (except for IgA-2) were higher in active UC than that found with remission. For CD, the percentage of IgG3-, IgG4-, and IgM-coated bacteria were higher than seen in the control population, and each was directly correlated with activity of disease.

In a third IBD study, the IgA-biome of 27 patients with CD, 8 patients with UC, and 20 healthy controls were determined using IgA-Seq methods [34]. Thirty-five intestinal bacterial taxa were found highly coated by IgA in CD and UC but not in the healthy controls. While strains of *Streptococcus luteciae*, *Hemophilus parainfluenzae,* and *Collinsella aerofaciens* were identified in both IBD and controls, they were only IgA-coated in the patients with IBD. Some IgA-coated species were uniquely associated with CD, including *Bulleidia*, *Allobaculum* spp., *Lactobacillus mucosae*, *Clostridiales*, *Ruminococcaceae* and *Blautia* spp. and others were found only in patients with UC, including *Eubacterium dolichum* and *Eggerthella lenta*. Strains of IgA+ bacteria from patients with IBD were then studied in germ-free mice where the strains elicited concordant IgA-coated bacterial responses in the mice and rendered them susceptible to colitis in the dextran-sodium-sulfate (DDS) model. 

Investigators in China, determined that the proportion of IgA/IgG-coated bacteria correlated directly with severity of patients with IBD [35].

In another study, binding of bacteria to IgA subclasses in patients with IBD was examined to see if the bacteria could contribute to the dysbiosis in these conditions [36]. The study found IgA-bound bacteria in CD and UC showed distinct IgA1- and IgA2-associated microbiota. The IgA1 in CD was enriched with potentially beneficial microbiota while the IgA-coated bacteria in UC were not. The authors conclude IgA-coating of bacteria in IBD could contribute to the dysbiosis seen in the disorders.

A separate study examined IgA-coated intestinal microbiota in 184 patients with IBD and 32 healthy controls [37]. Forty-three bacterial strains showed higher IgA coating in IBD than controls. Treatment with TNF-alpha altered the bacteria-specific IgA coating with a pattern correlating with clinical response to treatment. The authors conclude IgA responses to microbiota may serve as biomarkers of disease useful in assessing response to treatment.

In the next study, 55 patients with IBD (both CD and UC) and 50 healthy controls were studied for IgG-coating of bacteria [38]. The IgG-coated bacterial population showed a significantly lower microbiota diversity in the patients with IBD compared with controls. No differences were seen between CD and UC with their IgG-coated bacteria or between active and inactive disease.

The seven studies of IBD all show a central theme that IgA- or IgG-coated bacteria are associated with IBD pathogenesis and these coated strains elicit colitis in mice. It would be premature to assume the Ig-coating of the bacteria enhanced virulence as discussed later.

In Table 1A, a summary of taxa identified in the seven studies of the IgA-biome taxa associated with IBD is provided.

A complication of CD is peripheral spondyloarthritis. Viladomiu et al. [45] demonstrated in a study of 59 patients with IBD, with or without associated spondyloarthritis, that an IgA-coated *E. coli* enrichment was identified in the group with this rheumatologic complication. They then obtained strains from three patients and studied them for colonization of Caco-2 monolayers. One strain was selected as it adhered to the monolayer but did not penetrate the mucus. The strain was then tested in a germ-free mouse model and shown to colonize the gut and lead to a rise in IL-17 + CD4+ T cells not observed in controls. The authors conclude that IgA-Seq is a powerful tool capable of identifying immune-reactive pathosymbionts that link mucosal with systemic Th17-associated inflammation and suggest this approach may have therapeutic implications in Crohn’s disease and its complications.

### 7.2. Enteric Infection

IgA binds to pathogens like *Vibrio cholerae* or noroviruses during infection. In cholera, IgA-bound organisms have reduced ability to colonize the intestine or penetrate the epithelial mucus layer [46]. In norovirus infection, the process of attachment of the virus to cellular glycans, the initial step in attachment before penetration of the epithelial lining, is blocked by specific IgA [47]. Norovirus strains are capable of evading IgA-induced neutralization through mutation and antigenic variation or by alteration of virus-binding sites. Vaccines aimed at protection from enteric infection work by producing IgA that coats the bacteria or toxins produced, leading to pathogen neutralization (immune exclusion) [48]. 

### 7.3. Celiac Disease in Children

Celiac disease is a genetic autoimmune disorder where patients are intolerant to gluten found in foods, including wheat, rye and barely. In a study of pediatric celiac disease, the relative proportion of IgA-coated bacteria was significantly lower in 24 untreated children (*p* = 0.018) and 18 treated children (*p* = 0.003) compared with 20 healthy controls [39]. The proportions of IgG- and IgM-coated bacteria were also significantly lower in the treated celiac disease patients compared with in untreated celiac disease patients and controls (*p* = 0.009 and *p* < 0.001, respectively).

### 7.4. Childhood Allergies and Asthma

Twenty of 188 infants developing allergic symptoms and asthma during the first seven years of life were studied for IgA-coated fecal bacteria during the first year of life [41]. The proportions of IgA-coated fecal bacteria were significantly lower in these children developing allergic symptoms than in healthy controls. In the children developing allergies, *Fecalibacterium* spp. and *Bacteroides* spp. were largely unbound to IgA while being largely bound to IgA in the healthy control children.

In another study, 40 school age children with a history of asthma and 40 control children 9–17 years of age were studied for fecal and nasal IgA-coated bacteria [40]. The number of strains of fecal IgA-bound bacteria was decreased in the population with asthma compared with controls. Loss of IgA binding to strains of *Ruminococcus* genus and Lachnospiraceae (*Blautia*) were seen in the asthmatic group and reduced IgA-binding to fecal bacteria correlated with more severe asthma.

### 7.5. Undernutrition

Undernutrition occurs where enteric infections are common secondary to reduced hygiene standards. In a study of Malawian twin children discordant for kwashiorkor, the children with malnutrition were found to be colonized by disease-promoting enteric pathogens and IgA-coated Enterobacteriaceae (dysbiosis) that correlated with nutritional status and anthropometric measurements [49]. When gnotobiotic mice were administered IgA-bound strains of Enterobacteriaceae from the undernourished twin, weight loss and disrupted gut barrier with sepsis resulted. This was prevented by administering two IgA-coated bacteria from the properly nourished twin. The study demonstrated proinflammatory bacteria-coated by IgA can be associated with the development of malnutrition while enteric microbiota can prevent this complication.

In a second study of childhood nutrition-microbiome interactions in various African countries, 188 children 2–5 years of age, including 98 who were stunted, were screened for IgA-coated bacteria [50]. Stunted children had a greater proportion of IgA-coated bacteria compared to non-stunted controls (*p* = 0.029) with the two most commonly bound strains being pathogenic bacteria, *Campylobacter* and *Hemophilus.*

In the third experiment, mice exposed to diet-producing undernutrition developed a reduction in IgA-coating of *Lactobacillus* spp., independent of available levels of free IgA in the mice [51]. The study found glycan-mediated interactions between *Lactobacillus* and IgA antibodies were lost and leading to reduced mucosal colonization. In properly fed mice, IgA binding to *Lactobacillus* occurred.

### 7.6. Obesity and Type 2 Diabetes 

There are many studies showing obesity and type 2 diabetes are associated with dysbiosis and intestinal and systemic inflammation. The studies described below for obesity and type 2 diabetes, as well as those above dealing with inflammatory bowel disease, did not find a beneficial role of the IgA-biome in the conditions. 

In the first study, fecal immunoglobulin-coated bacteria were studied in 40 obese patients with type 2 diabetes and 40 patients with obesity without diabetes before and after bariatric surgery [42]. Body weight, fasting glucose levels and markers of inflammation decreased post-surgery while lipopolysaccharide (LPS)- and flagellin-specific IgA and pro-inflammatory strains of IgA-coated Enterobacteriaceae increased in feces. Overall increased IgA-coated bacteria post-surgery was not seen. The authors conclude that post bariatric surgery there is an expansion of proinflammatory bacteria that is compensated by an improved IgA antibody response that has beneficial effects by reducing systemic inflammation.

In a study of 8 diabetics, 8 prediabetics and 8 non-diabetics based on fasting blood glucose value, fecal levels of IgA- and IgM-coated bacteria were determined [52]. The percent of IgA/IgM-coated bacteria for the diabetic, prediabetic and non-diabetic were 12.23%, 10.63%, and 11.1%, respectively. The IgA-biome as a group showed greater differences in salivary secretions than in feces. IgA-bound organisms identified in the diabetic group were known to commonly be part of the intestinal microbiome of diabetics. 

The next investigators looking at obesity and type 2 diabetes designed studies to determine if IgM antibodies to gut microbiota may play a role in the pathogenesis of obesity and type 2 diabetes in both animal and human subjects. The study used a breed of germ-free mice known to secrete only fecal IgM-coated bacteria. They also collected human fecal samples from children and adolescents with obesity and glucose-tolerance, -intolerance or frank diabetes for use in later fecal microbiota transplantation studies [53]. The strain of germ-free mice was given a high fat diet to induce obesity and their fecal pellet was infused into wild-type mice resulting in weight gain. The fecal product from obese children with type 2 diabetes were shown to possess a high concentration of IgM-coated bacteria. Using their stools as fecal transplants into wild strain mice, weight gain and glucose intolerance resulted. The authors conclude that IgM-bound bacteria play a potential role in the immunopathogenesis of obesity and type 2 diabetes.

### 7.7. Clostridioides Difficile Infection (CDI)

In CDI, the extreme dysbiosis that exists because of receipt of multiple antibiotics dominates microbiome findings, and the IgA-biome is similarly impaired. 

In a study of active CDI in 24 patients, the infecting strains of *C. difficile* were the dominant fraction of IgA-coated bacteria, while strains belonging to *Clostridium* cluster IV (*Eubacterium*, *Ruminococcus* and *Anaerofilum* genera) and Lactobacillales order showed decreased IgA-coating [54]. In the uninfected healthy control group, *Fusobacterium* was the dominate IgA-coated bacterium. 

In a study of 48 patients with recurrent CDI, strains of pro-inflammatory Enterobacteriaceae were the most highly IgA-coated [55]. The IgA-bound bacterial population showed a reduced number of operational taxonomic units (OTUs) (alpha diversity) and reduced number of specific taxa (beta diversity) in the group with recurrent CDI compared with stools obtained from the subjects after transplantation and with stools from the healthy donors providing the fecal microbiota transplant product.

### 7.8. Irritable Bowel Syndrome (IBS)

IBS shows clinical overlap with IBD. While inflammation plays a role in disease pathogenesis in IBS [56], it is at a much lower level than in IBD and the mucosa is histologically normal. 

Forty-four patients with diarrhea-predominant irritable bowel syndrome (IBS-D) and 32 healthy volunteers were studied for the presence of IgA-coated bacteria [57]. The patients with IBS-D showed a higher abundance of IgA-coated bacteria compared with controls (*p* = 0.0024). Three highly IgA-coated taxa were strongly associated with IBS-D compared with controls: Enterobacteriaceae, *Granulicatella*, and *Hemophilus*. Abundance of Enterobacteriaceae strains was correlated also with anxiety and depression. The authors concluded microbial dysbiosis promote higher levels of IgA in the terminal ileum, increasing concentration of IgA-coated bacteria. The authors question if IgA may have secondary effects on microbial dysbiosis in IBS-D.

Using a murine restraint model of stress, an animal model of IBS, a study provided evidence that stress produced diarrhea, dysbiosis, enhanced IgA-binding to bacteria, and bacterial translocation by opening of colonic goblet cell-associated passages (GAPs) [58].

### 7.9. Multiple Sclerosis (MS)

Neurodegenerative disorders, including MS, Parkinson’s disease and Alzheimer’s disease all are associated with dysbiosis with evidence that gut microbiota play a role in disease pathogenesis. 

In a study designed to examine the relationship between immune coated bacteria in the disease, 36 patients with remission or relapsing MS and 31 healthy controls were enrolled in a study of IgA-coated bacteria [59]. The proportion of IgA-bound bacteria (OTUs) were higher in patients with MS compared with the controls. The investigators then demonstrated that IgA-B cells in the central nervous system associated with neuroinflammation cross-reacted with surface structures on bacterial strains from the gut. 

In a second study of MS, 30 patients with relapsing-remitting MS and 32 healthy donors were studied for commensal-specific antibody responses [43]. Bacteria strains bound to IgA were significantly reduced in patients with severe multiple sclerosis compared with controls (*p* < 0.0001). IgA-unbound bacteria were associated with an increased serum IgG response in MS patients compared with controls. The authors conclude a defective IgA response in MS leads to a systemic IgG response against enteric microbiota early in MS that could play a role in pathogenesis of the disease.

In a mouse model of MS, gut-derived IgA antibody-secreting plasma cells that had migrated to the central nervous system were shown to control clinical disease in an IL-10-dependent manner [60]. Mice with more active MS demonstrated an increased infiltration of commensal-specific IgA antibody producing plasma cells in the CSF, which the authors felt could be a marker of acute inflammation in the disease. The study then looked at IgA-binding to fecal bacteria that were found to be lower in frequency during active disease than seen for controls. Mice with actively relapsing disease had significantly reduced IgA-bound gut bacteria compared with those in remission, thought by the authors to represent the movement of IgA-producing cells to the brain from the intestine during periods of inflammation. 

### 7.10. Breast Cancer

Forty-eight post-menopausal women with breast cancer were found to have dysbiosis, with an IgA-biome showing reduced alpha diversity (*p* = 0.012) and unique beta diversity bacterial composition (*p* = 0.02) compared with 48 aged-matched controls [44]. These changes in their microbiome, both IgA-biome and IgA-negative population correlated with levels of urinary estrogen. Cases were more likely to be colonized by IgA-coated *Ruminococcus oscilibacter* than controls (*p* = 0.003).

Table 1B includes a summary of IgA-coated bacteria taxa identified in the studies cited above, excluding inflammatory bowel disease, which is summarized in Table 1A. Because each study provides different depth on identification of organisms, from phyla to genus, complete taxonomy of the IgA-coated bacteria identified in the study is provided. To be included in this list a taxon needed to be mentioned in two or more of seven studies.

An integrated summary of the IgA-biome studies discussed above (Section 7.1, Section 7.2, Section 7.3, Section 7.4, Section 7.5, Section 7.6, Section 7.7, Section 7.8, Section 7.9 and Section 7.10) is included in Table 2.

### 7.11. Summary of Targets and Effects of the Immune-biome in Disorders of Dysbiosis

In Figure 2 we summarize four scenarios wherein the immune-biome is associated with conditions associated with dysbiosis and compare them with the IgA-biome in healthy people. In the first, with inflammatory bowel disease, the IgA- or IgG-coated bacteria commonly are strains of proinflammatory bacteria likely to be involved in the pathogenesis of the disease. 

In the second setting of enteric infection caused by an infectious agent (e.g., *V. cholerae*, *C. difficile* or norovirus), IgA-coating of the pathogenic organism occurs as a body defense mechanism in attempting to control or eliminate the infection. 

In the third where the intestinal mucosa is histologically normal, but inflammation occurs, IgA- or IgM-coated bacteria are elicited presumably as a secondary response to the inflammatory process. 

In the fourth, chronic disease states are associated with reduction in IgA-biome, contributing to the overall dysbiosis. 

## 8. IgA-Biome before and after FMT in Disorders Associated with Dysbiosis 

### 8.1. Inflammatory Bowel Disease

Twelve patients with UC received a course of antibiotics and then were randomized to receive 12 weeks of FMT by capsules or placebo with extensive characterization of their fecal microbiome [61]. FMT shifted the IgA enrichment of bacteria to that seen in the donor. Engraftment dynamics were followed over time in the FMT group, documenting long term persistence of some transplanted strains in the recipients while others never engrafted. The study demonstrated that phylogenetically diverse IgA-coated bacteria could be transferred from donors to recipients without losing their immune-bound state. 

Twenty patients with ulcerative colitis were given FMT product from 2 donors and subsequently studied for microbiome changes by deep shotgun metagenomic sequencing and by IgA-Seq [62]. The proportion of fecal bacteria coated with IgA were equivalent pre- and post-FMT in clinical responders and clinical non-responders, while the IgA-biome four weeks after FMT was more diverse, showing increase in alpha diversity by Shannon index and beta diversity by unweighted UniFrac analysis, compared with pre-treatment studies. The improved diversity correlated positively with clinical response. Twenty-nine IgA-coated bacterial communities were shared by donors and FMT recipients: *Bacteroides* (5 strains), *Coprococcus* (2 strains), *Eubacterium* (1 strain)*,* Lachnospiraceae (4 strains), *Odoribacter* (1strain), and *Ruminococcus* (3 strains). These IgA-coated taxa were subsequently transferred to germ-free mice and studied. Eleven donor derived IgA-coated taxa were found in the transplanted mice, including strains of *Allistipes*, *Odoribacter*, and *Ruminococcus*. Of the donor taxa, only *Osoribacter splanchnicus* was significantly correlated with clinical improvement. 

Sixty-one patients with IBD were studied for IgA- and IgG-coated bacteria with 18 subjects consenting to FMT [35]. In the 18 patients receiving FMT, sIgA-/IgG-biomes were studied before and after FMT. Strains identified as *Escherichia coli-Shigella* (Proteobacteria phylum) were shown to be highly bound to IgA before FMT. The 18 patients who underwent FMT showed no change in percentage of IgA/G-bound bacteria before versus after FMT although the absolute number of IgA/G-bound bacteria decreased as well as level of free fecal IgA and IgG. The alpha diversity of the microbiome increased after FMT as determined by Choa and Shannon indexes, and the principal coordinates analysis (PCoA) showed extensive changes in IgA-biome before and after FMT. At the genus level, the abundance of IgA targeted taxa after FMT that were increased included UCG-002, *Agathobacter*, *Ruminoococcus_torques*_group, and *Subdoligranulum*, while *Serratia* decreased. Route of FMT (gastric or colonoscopy) did not influence the pattern of IgA-/IgG-coating in the 18 subjects with IBD. Finally, in the study the investigators demonstrated the elevated IgA/G+ bacteria found in a DDS mouse model of colitis at baseline was reduced to levels seen with control mice by treatment with FMT. 

Determinants of resistance to dextran-sodium-sulfate-induced colitis were studied in wild-type inbred strains of B6 and CBA mice that differed in the amount of IgA that they made but were otherwise healthy [63]. Resistance to DDS colitis was seen in the mice with higher IgA-biome. In mice susceptible to the DDS colitis model, administration of a fecal transplant high in IgA-coated bacteria protected the mice. The study reported that high-IgA-coating of bacteria found in older mice can protect mice early in life against colitis when they are most susceptible. 

### 8.2. Recurrent Clostridioides Difficile Infection (CDI)

Forty-eight patients with recurrent CDI were studied for IgA-bound bacteria before and after FMT [55]. Prior to FMT, *E. coli* (Enterobacteriaceae family) was the most highly IgA-bound taxon, followed by bound strains in the Firmicute phylum. In successful response to FMT, the IgA-coating of recipients mirrored the findings seen in the donors’ microbiome, unaffected by route of FMT (oral capsules or colonoscopy). 

### 8.3. Systemic Sclerosis

Systemic sclerosis, an auto-immune, multi-system disorder, is known to be associated with reduced microbiome diversity. In an FMT treatment of study patients with systemic sclerosis, five were randomized to receive FMT and four to a placebo [64]. The microbiomes of the two groups were equivalent at baseline. At 16 weeks, but not at 4 weeks, differences in fecal microbiota were identified with improvement in alpha diversity (*p* < 0.006), beta diversity (*p* < 0.02), and changes in abundance of IgA- and IgM-coated bacteria in the group receiving FMT that was associated with improvement in GI symptoms. 

### 8.4. Aging of Germinal Centers and Reduction of Intestinal IgA Produced

It is well known from multiple studies that the composition and diversity of the gut microbiome reduces with aging. Mice with advanced age (22 months old) were examined for reactivity of IgA production sites [65]. Germinal centers (GC) and Peyer’s patches (PP) before and after receipt of multiple transplantations with fecal pellets from adult (3-months old) mice were examined. The study mice were also housed with younger aged mice and given a dose of cholera toxin to stimulate an immune response. GC reactivity was improved by FMT that was boosted by the vaccine. The defective GC capacity in older mice was shown to be reversible with fecal transplantation and vaccine administration with enhancement of immune (IgA) reactivity to commensal bacteria when comparing findings before and after treatment.

An integrated summary of the FMT studies discussed above (Section 8.1, Section 8.2, Section 8.3 and Section 8.4) is included in Table 3.

**Table 3 microorganisms-11-00093-t003:** IgA-biome in Conditions Associated with Dysbiosis Treated with Fecal Microbiota Transplantation.

Condition	Findings	Comment
IBD	The percent of IgA/IgG-coated bacteria was elevated in IBD. FMT shifted the IgA-enriched bacteria to that of the donors [61]. Route of FMT (oral or colonoscopy) did not influence the pattern of IgA/IgG-coated bacteria. Diversity of the IgA-biome at 4 weeks after FMT was more diverse and the diversity correlated with clinical response in UC [62]. Of the donor taxa only *Osoribacter splanchnicus* was significantly correlated with clinical improvement in one study [62]. The IgA-biome is effective in preventing colitis in mice exposed to dextran-sodium-sulfate (DDS), an IBD model [32,63].	FMT shifted the IgA enrichment of bacteria to that of the donor in patients with UC. Phylogenetically diverse IgA-coated bacteria were transferred from donors to recipients. Diversity of the IgA-biome four weeks after FMT correlated with clinical response.
Recurrent CDI	Proinflammatory Proteobacteria strains were the most highly bound to IgA. Post-FMT, the IgA-biome of patients mirrored the donors [55].	The dysbiosis in these patients relates to multiple antibiotics received. Replacing the microbiota with healthy taxa coated by IgA is curative in most cases.
Systemic Sclerosis	Abundance, richness and diversity of IgA-coated and IgM-coated bacteria fluctuated dynamically after transplantation in the FMT group, not in the placebo, and GI symptoms were improved [64].	More studies are needed with systemic sclerosis to determine the value of FMT in disease management.
Aging	Germinal centers (GC), important in the production of IgA, lose functional capacity with aging. Advanced age mice, treated with FMT and cholera toxin vaccine experienced an increase in functional GCs [65].	Impaired immune (IgA) reactivity of gut associated lymphoid tissues (GALT) in aging mice was shown to be reversible by FMT from younger aged donors.

## 9. Summary and Comments

While it has been known for some time that secretory IgA was the most abundant immunoglobulin in the body playing a central role in preventing and treating infections by enteric pathogens, the role of IgA-biome in gut health and predictive value in FMT are areas of emerging science. In this review we described the known biologic activity of the IgA-biome and then looked at a group of disorders associated with microbiome states in disarray where antibody coated-bacteria were studied to determine association with disease. 

### 9.1. Five Scenarios May Best Explain the Biology of the IgA-Biome in Different Clinical Settings

In drawing conclusions from the data available on the IgA-Biome, there are five clinical scenarios that may help explain the published findings. First, for healthy people the IgA-biome plays a beneficial homeostatic role in the gastrointestinal tract and should provide predictive value post FMT. The importance of this to overall health remains undefined. The second clinical setting involves patients with inflammatory bowel disease where IgA is found to coat strains of proinflammatory Enterobacteriaceae that are involved with disease pathogenesis. The third is seen with infection by a frank pathogen or colonization by proinflammatory bacteria that lead to coating of the infecting or colonizing strains by available IgA as the body’s attempt to minimize pathogen growth and damage. The fourth category is seen with focal or systemic inflammation without mucosal damage as seen in irritable bowel syndrome, and type 2 diabetes with obesity and certain rheumatologic disorders. The final group is defined by depressed levels and diversity of the IgA-biome related to chronic diseases showing decreased IgA production [66], absence or depletion of the critical bacterial strains that stimulate release of IgA [67], or destruction of the immune cells [68]. 

Investigators performing studies in patients with inflammatory bowel disease and obesity with type 2 diabetes implied that the IgA-coating of proinflammatory bacterial strains contributed to disease pathogenesis. A more likely explanation is that virulent or proinflammatory strains that do contribute to the pathogenesis of the inflammatory disorders elicit antibodies as a host response to control the infection. While unproven, the IgA coating of these pathogenic bacteria may lead to organism attenuation. Consistent with this theory, and as mentioned above, immune-coated bacteria have predictive value in successful FMT in inflammatory bowel disease. 

### 9.2. New Studies Suggested Evaluating IgA-Biome in Fecal Microbiota Transplantation

In addition to the need for increased research on the role of IgA-biome in health and disease and predictive value in FMT, specific projects should be carried out to better define its biology and importance. The first is to compare mucosal associated microbiota with gut luminal strains for IgA-coating. Most studies have used fecal samples as representative of colonic microbiota. A more fruitful approach to study the immune microbiome would be to obtain mucosal biopsies [69], endoscopically collected lavage [70] or mucosal cultures through a protected device [71] to obtain bacteria present in the mucus layer. Bacteria associated with the mucosa differ from those in the free lumen and may be the ideal source for finding the important strains of the IgA-biome. As part of this line of research, studying changes in mucosal associated- and luminal-bacteria in patients being treated with FMT for dysbiosis may yield more clinically relevant data on microbiome recovery [72]. 

More work is needed to clarify the role of the IgA-biome in FMT. It should be measured in both donors and recipients, and engraftment into the recipients should be followed, correlating this with recovery of the microbiome. Studies cited above document the transfer of IgA-coated bacteria from donors to recipients. Chu et al. [61] followed IgA-coated bacteria that remained strongly coated through time when transferred across hosts from donors to recipients. These investigators found that unique bacterial strains transferred between the donor and recipient retained their IgA coating while strains shared between the two were less likely to be IgA-coated suggesting that IgA-coated bacteria are unique and identifiable in the intestine across individuals.

A third line of research that would help define the dynamics of microbial restoration through fecal microbiota transplantation is to expand the studies of the IgA-biome into a broader array of non-bacterial microbiota. More work is needed to study immune response to gut viruses and possibly fungi. Wherever there are bacteria in the body, there are viruses. Bacteriophages in the gut that infect bacteria contribute to bacterial function, transfer of genes and organism survival [73], and can contribute to the development of dysbiosis [74]. The mucosal surface contains up to 10^9^ phages per biopsy [75] and elicit antibodies that promote their uptake and clearance [76]. Antiphage antibodies can be IgA, IgM or IgG, although IgM may be as important as IgA or more important in this context [77]. Finding that bacteria-free fecal product successfully led to improved microbiome and cure in patients with recurrent *C. difficile* infection [78], supports the idea viruses or possibly metabolites are important mediators in FMT.

The intestinal mycobiome is largely unstudied but is involved with health and disease [79]. Intestinal fungi do elicit an IgA response in the gut and can be IgA-coated potentially influencing its biology in important ways [80].

### 9.3. Review Limitations

The review has limitations. First, the number of studies on the IgA-biome are limited. Secondly, the state of metagenomics is an evolving science. In the studies that identify IgA-coated bacteria the publications use designations that range from phyla to genera. Deeper sequencing will provide a better understanding of the microbiome where all investigators working in this field are able to identify microbiota to the species or even strain levels [81].

## Figures and Tables

**Figure 1 microorganisms-11-00093-f001:**
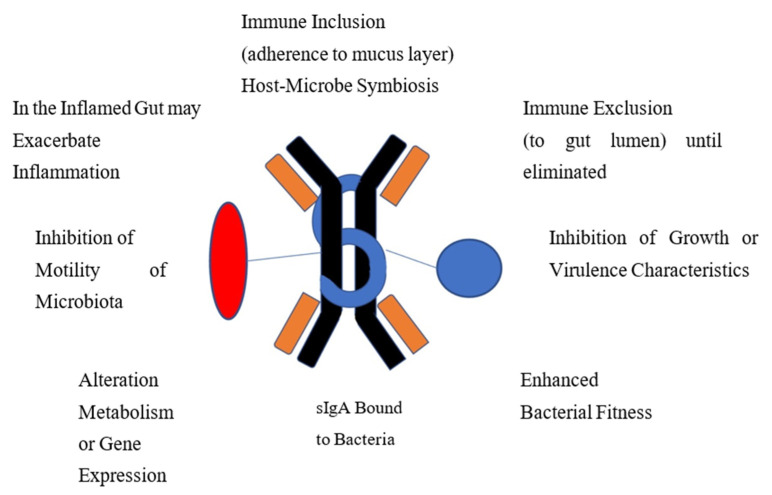
Proposed Mechanisms Whereby Immune-Coated Bacteria Functions to Provide Homeostasis and Gut Health.

**Figure 2 microorganisms-11-00093-f002:**
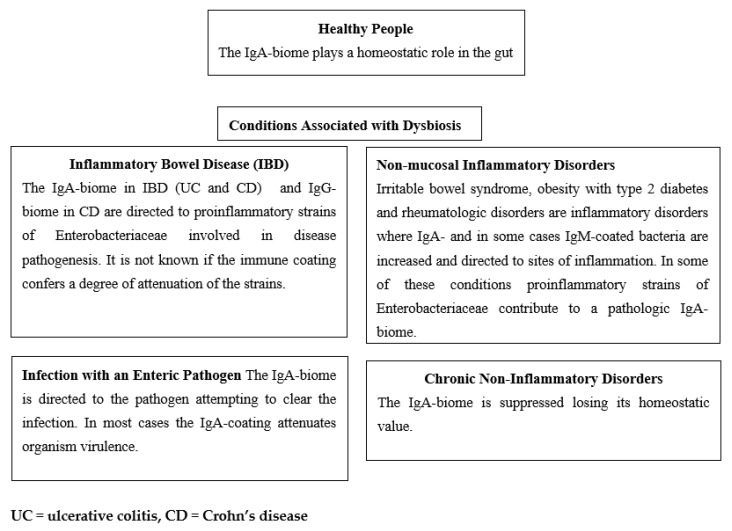
Targets and Effects of Immune-biome in Healthy People and in Disorders with Dysbiosis.

**Table 1 microorganisms-11-00093-t001:** Ig-Coated Bacterial Taxa Identified in Published Studies Employing Sequencing Methods. To be Included an Organism Needed to be Identified in Two Separate Studies. (**A**). Taxa Identified in Studies of Patients with Inflammatory Bowel Disease [32,33,34,35,36,37,38]. (**B**)**.** Taxa Identified in Studies of Patients with Dysbiosis Associated with a Variety of Other Causes [16,39,40,41,42,43,44].

(A)				
Phylum	Order	Family	Genus	Species
Firmicutes (Bacillota)				
	Clostrdiales	Lachnospiraceae	*Roseburia*	Roseburia spp.
	Eubacteriales	Clostridiaceae	*Clostridium*	*Clostridium* spp.
	Eubacteriales	Oscillospiraceae	*Ruminococcus*	*Ruminococcus* spp.
	Eubacteriales	Oscillospiraceae	*Faecalibacterium*	*Faecalibacterium prausnitzii*
	Eubacteriales	Lachnospiraceae	*Blautia*	*Blautia* spp.
	Eubacteriales	Lachnospiraceae	*Coprococcus*	*Coprococcus* spp.
	Eubacteriales	Lachnospiraceae	*Anaerostipes*	*Anaerostipes* spp.
	Lactobacillales	Streptococcaceae	*Streptococcus*	*Streptococcus* spp.
	Vellionellales	Veillonellaceae	*Veillonella*	*Veillonella* spp.
	Vellionellales	Veillonellaceae	*Dialister*	*Dialister* spp.
	Erysipelotrichia	Turicibacteraceae	*Turicibacter*	*Turicibacter* spp.
Bacteroidetes (Bacteroidota)				
	Bacteroidales	Bacteroidaceae	*Bacteroides*	*Bacteroides ovatus*
	Bacteroidales	Rikenellaceae	Too many to list	Too many to list
Proteobacteria (Pseudomonadota)				
	Enterobacterales	Enterobacteriaceae	Too many to list	*E. coli*, *Klebsiella*, etc.
	Pasteurellales	Pasteurellaceae	*Hemophilus*	*Hemophilus* spp.
Actinobacteria (Actinomycetota)				
	Coriobacteriales	Coriobacteriaceae	*Collinsella*	*Collinsella* spp.
	Bifidobacteriales	Bifidobacteriaceae	*Bifidobacterium*	*Bifidobacterium* spp.
	Eggerthellales	Eggerthellaceae	*Eggerthella*	*Eggerthella* spp.
**(B)**				
**Phylum**	**Order**	**Family**	**Genus**	**Species**
Firmicutes (Bacillota)				
	Clostrdiales	Lachnospiraceae	*Roseburia*	*Roseburia* spp.
	Lactobacillales	Enterococcaceae	*Enterococcus*	*Enterococcus* spp.
	Eubacteriales	Clostridiaceae	*Clostridium*	*Clostridium* spp.
	Eubacteriales	Oscillospiraceae	*Ruminococcus*	*Ruminococcus* spp.
	Eubacteriales	Oscillospiraceae	*Faecalibacterium*	*Faecalibacterium prausnitzii*
	Eubacteriales	Lachnospiraceae	*Blautia*	*Blautia* spp.
	Eubacteriales	Lachnospiraceae	*Coprococcus*	*Coprococcus* spp.
	Eubacteriales	Lachnospiraceae	*Anaerostipes*	*Anaerostipes* spp.
	Clostridiales	Eubacteriaceae	*Eubacterium*	*Eubacterium* spp.
	Erysipeiotrichia	Erysipelotrichaceae	Too many to list	Too many to list
	Lactobacillales	Lactobacillaceae	*Lactobacillus*	*Lactobacillus* spp.
	Lactobacillales	Streptococcaceae	*Streptococcus*	*Streptococcus* spp.
Bacteroidetes (Bacteroidota)				
	Bacteroidales	Bacteroidaceae	*Bacteroides*	*Bacteroides* spp.
	Bacteroidales	Prevotellaceae	*Prevotella*	*Prevotella* spp.
Verrucomicrobia (Verrucomicrobiota)				
	Verrucomicrobiales	Akkermansiaceae	*Akkermansia*	*Akkermansia muciniphila*
Proteobacteria (Pseudomonadota)				
	Pseudomonadales	Pseudomonadaceae	*Pseudomonas*	*Pseudomonas* spp.
	Enterobacterales	Enterobacteriaceae	53 different genera	*E. coli*, *Klebsiella*, etc.
	Burkholderiales	Alcaligenaceae	Too many to list	Too many to list
Actinobacteria (Actinomycetota)				
	Coriobacteriales	Coriobacteriaceae	*Collinsella*	*Collinsella* spp.
	Bifidobacteriales	Bifidobacteriaceae	*Bifidobacterium*	*Bifidobacterium* spp.
	Eggerthellales	Eggerthellaceae	*Eggerthella*	*Eggerthella* spp.
	Propionibacteriales	Nocardioidaceae	*Aeromicrobium, Marmoricola, Mumia* and *Nocardioides*	More than 50 species

**Table 2 microorganisms-11-00093-t002:** IgA-Coated Bacteria in Conditions Associated with Dysbiosis Which Represent Target Conditions for Microbial Restoration Through Fecal Microbiota Transplantation, Summary of Findings.

Condition	General Findings	Comment
Inflammatory Bowel Disease	Ig-coated bacteria were increased in IBD [32,33,34,35,36,37] and correlated directly with inflammation markers and severity of disease and facilitated development of colitis in mice [34]. The diversity of the Ig-coated population was lower than the core Ig-negative microbial population [38]. Treatment with TNF-alpha was associated with a change in pattern of IgA-coated bacteria that predicted response to treatment [37].	The enhanced production of Ig appears to be a direct result of pathogenic strains of bacteria producing inflammation that are involved in disease pathogenesis. Changes in coated IgA-bacteria may provide predictive value for therapeutic response in IBD.
Enteric Infections	IgA binds to viral, bacterial and parasitic pathogens and influences growth and virulence of the strains [46,47,48].	This is an example of “immune exclusion”.
Celiac Disease in Children	IgA-, IgG- and IgM-coated bacterial levels were low in childhood celiac disease, treated or untreated [39].	Gluten-free microbiome friendly diet may improve the microbiome in these children.
Childhood Allergies and Asthma	IgA-coated fecal bacteria were reduced in asthma with the level of reduction corelating with more severe disease [40,41].	Low Ig-coated bacteria impairs gut health, and a microbiome friendly diet should be evaluated in these children.
Undernutrition	IgA-coated fecal bacteria were reduced [51] unless infected by enteric pathogens or colonized by pro-inflammatory strains of Enterobacteriaceae [49,50]	Low Ig-coated bacteria impairs gut health, protein caloric intake and microbiome friendly diets are needed.
Obesity and type 2 diabetes	In a mouse model of obesity, IgM-coated bacteria appeared to be involved in the immunopathogenesis of obesity and type 2 diabetes [53]. In humans with diabetes, IgA commonly coats strains of proinflammatory Enterobacteriaceae that appear to contribute to regulation of obesity-related insulin resistance [42].	The microbiome and Ig-coated bacteria need more study in obesity and diabetes with potential improvement in microbiome diversity with FMT.
*C difficile* Infection	The total IgA-microbiome was largely depleted in this infection, with proinflammatory strains of IgA-coated Enterobacteriaceae dominating the microbiome [55].	Antibiotic effects on normally protective microbiota are the driving force leading to dysbiosis and susceptibility to CDI.
Irritable Bowel Syndrome	The proportion of IgA-coated bacteria was elevated in IBS-D, was associated with emergence of proinflammatory taxa [57], and in an animal model enhanced enteric symptoms and potentiated bacterial translocation [58].	Patients with IBS-D show increased proportions of IgA-coated bacteria that may be involved with the pathogenesis of the disease and may represent therapeutic targets.
Multiple Sclerosis	IgA-coated bacteria were reduced in multiple sclerosis [43,60].	Defective IgA responses in MS need further study and the authors suggest that microbiota reconstitution with fecal microbiota transplantation should be evaluated in MS to look for clinical and biologic effects.
Breast Cancer	IgA-coated bacteria were reduced in women with breast cancer [44] and the IgA-microbiome appeared unique, with increased proportion of IgA-coated *Ruminococcus oscilibacter* (*p* = 0.003). The microbiome changes seen correlated with urinary estrogen levels.	More studies are needed of the Ig-biome in cancer.

## Data Availability

This is a review article and all data reviewed are available in the publication.

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
