# Peer review of "Intestinal IgA-Coated Bacteria in Healthy- and Altered-Microbiomes (Dysbiosis) and Predictive Value in Successful Fecal Microbiota Transplantation"

_microorganisms, 2022, doi:10.3390/microorganisms11010093_

Round 1

Reviewer 1 Report

This review entitled “Intestinal IgA-Coated Bacteria in a Healthy- and an Altered-Microbiomes (Dysbiosis) and Predictive Value in Successful Fecal Microbiota Transplantation” is very interesting and novel, since relevant studies are limited. The authors gave a nice aspect of view on this topic, which will be the basis for further relevant research. It is a well written paper, presented with a comprehensive way. The conclusion is consistent with the evidence and arguments presented. Despite the limitations reported, the review should be published. Therefore, I recommend to be accepted.

Author Response

Comments to reviewers

Reviewer 1

This review entitled “Intestinal IgA-Coated Bacteria in a Healthy- and an Altered-Microbiomes (Dysbiosis) and Predictive Value in Successful Fecal Microbiota Transplantation” is very interesting and novel, since relevant studies are limited. The authors gave a nice aspect of view on this topic, which will be the basis for further relevant research. It is a well written paper, presented with a comprehensive way. The conclusion is consistent with the evidence and arguments presented. Despite the limitations reported, the review should be published. Therefore, I recommend to be accepted.

Thank you for your comments.

Reviewer 2 Report

Authors DuPont et al. have conducted a scoping review IgA-biome and subsequent importance in the success of FMT. This is an important topic given the demonstrated efficacy of FMT and its notorious failure rate. Investigating the role of the IgA-biome is therefore a worthy topic to cover. However, there are some major improvements that can be easily made to the paper which would substantially improve its quality. Those are:

1. Please read through the paper very carefully to check for English issues, there are grammar issues from the abstract to the conclusion, please fix. Also, please make sure all genus and species names are italicized. And you don't have to capitalize biome in IgA-Biome, you should write it as IgA-biome in all instances. 

2. I understand this is not a systematic review, but the authors must indicate the methods they used to collate their selected articles. What databases did they search, and for what terms? Along what dimensions did they include or exclude articles? Please include a methods section. 

3. The authors should include a figure demonstrating the targets and effects of IgA-coated bacteria in disorders of dysbiosis. It would be great to see a biological schema demonstrating how IgA promotes gut diversity and how those biological mechanisms and underlying pathways are impacted. This can be Figure 2. 

4. Each finding summarized in Tables 1 and 3 should be cited. Therefore, every finding the authors claim in those columns can be quickly referenced. 

Following these major revisions, I recommend this paper to be accepted. I believe these revisions will significantly improve the quality and value of their paper. 

Author Response

Reviewer 2

Authors DuPont et al. have conducted a scoping review IgA-biome and subsequent importance in the success of FMT. This is an important topic given the demonstrated efficacy of FMT and its notorious failure rate. Investigating the role of the IgA-biome is therefore a worthy topic to cover. However, there are some major improvements that can be easily made to the paper which would substantially improve its quality. Those are:

  1. Please read through the paper very carefully to check for English issues, there are grammar issues from the abstract to the conclusion, please fix. Also, please make sure all genus and species names are italicized. And you don't have to capitalize biome in IgA-Biome, you should write it as IgA-biome in all instances. 

We went through the manuscript looking for typos. Thank you for pointing these out.

  1. I understand this is not a systematic review, but the authors must indicate the methods they used to collate their selected articles. What databases did they search, and for what terms? Along what dimensions did they include or exclude articles? Please include a methods section. 

We add Methods to the Introduction, Introduction and Methods, and then provided a new paragraph at the end:

In performing this review, the authors examined the literature by searching in PubMed for the following: immune microbiome, IgA AND microbiome, IgA coated intestinal microbiota, IgA AND inflammatory bowel disease, IgA AND dysbiosis, IgA AND inflammation. Additionally references identified led us to other published articles for review. (This is a new concept, requiring a variety of searchers as it went along. Searching for IgA-biome in Pub Med only identified two papers). We hope this publication will provide researchers interested in the microbiome with a current perspective and adequate bibliography on the topic. 

  1. The authors should include a figure demonstrating the targets and effects of IgA-coated bacteria in disorders of dysbiosis. It would be great to see a biological schema demonstrating how IgA promotes gut diversity and how those biological mechanisms and underlying pathways are impacted. This can be Figure 2. 

This was a good suggestion and is now included. See below.

  1. Each finding summarized in Tables 1 and 3 should be cited. Therefore, every finding the authors claim in those columns can be quickly referenced. 

Done. Thank you for the suggestion.

Figure 2. Targets and Effects of Immune-Biome in Disorders of Dysbiosis (attached as PDF)

Reviewer 3 Report

This is a very interesting review, but I have a few comments:

1) "The proportion of gram-positive bacteria vs. gram-negative and facultative vs. strictly anaerobic bacteria increases as we move from proximal to distal regions of the intestine." - the authors cite the error of the cited article. In fact, when moving from proximal to distal regions of the intestine, the proportion of facultative anaerobes decreases, and the proportion of strict anaerobes increases.

2) "Composition of bacteria vary as well with strains of several families including Streptococcaceae, Actinomycetaceae, Actinomycinaceae and Corynebacteriaceae are commonly found in the normal healthy small intestine, and strains belonging to the phyla or family of Firmicutes, Bacteroidota, and Actinomycetaceae are more commonly found in the colon." - Streptococcaceae is a family of the phylum Firmicutes. The sentence needs to be edited.

3) "Viladomiu et al [45] demonstrated in a study of 59 patients with IBD, with or without associated spondyloarthritis, that an IgA-coated E. coli enrichment was identified in the group with this orthopedic complication" - Spondyloarthritis is not an orthopedic but a rheumatological complication of IBD.

Author Response

Reviewer 3

This is a very interesting review, but I have a few comments:

  • "The proportion of gram-positive bacteria vs. gram-negative and facultative vs. strictly anaerobic bacteria increases as we move from proximal to distal regions of the intestine." - the authors cite the error of the cited article. In fact, when moving from proximal to distal regions of the intestine, the proportion of facultative anaerobes decreases, and the proportion of strict anaerobes increases.

This change was made.

  • "Composition of bacteria vary as well with strains of several families including Streptococcaceae, Actinomycetaceae, Actinomycinaceae and Corynebacteriaceae are commonly found in the normal healthy small intestine, and strains belonging to the phyla or family of Firmicutes, Bacteroidota, and Actinomycetaceae are more commonly found in the colon." - Streptococcaceae is a family of the phylum Firmicutes. The sentence needs to be edited.

We removed any comment on taxonomy. This now reads: Composition of bacteria vary as well with strains of Streptococcaceae, Actinomycetaceae, Actinomycinaceae and Corynebacteriaceae more commonly found in the normal healthy small intestine, and strains belonging to Firmicutes, Bacteroidota, and Actinomycetaceae more commonly found in the colon (20).

3) "Viladomiu et al [45] demonstrated in a study of 59 patients with IBD, with or without associated spondyloarthritis, that an IgA-coated E. coli enrichment was identified in the group with this orthopedic complication" - Spondyloarthritis is not an orthopedic but a rheumatological complication of IBD.

Changed, thank you.